

# Description of a new *Pangasius* (Valenciennes, 1840) species, from the Cauvery River extends distribution range of the genus up to South Western Ghats in peninsular India

Kathirvelpandian P.V. Ayyathurai[1], Paramasivam Kodeeswaran[1], Vindhya Mohindra[2], Rajeev K. Singh[2], Charan Ravi[1], Rahul Kumar[1], Basheer Saidmuhammed Valaparambil[1], Ajith Kumar Thipramalai Thangappan[1], Joykrushna Jena[3] and Kuldeep K. Lal[2]

[1] Peninsular and Marine Fish Genetic Resources Division, National Bureau of Fish Genetic Resources, Kochi, Kerala, India
[2] Fish Exploration and Conservation Division, National Bureau of Fish Genetic Resources, Lucknow, Uttar Pradesh, India
[3] Fisheries Division, Indian Council of Agricultural Research, New Delhi, India

Corresponding author
Kuldeep K. Lal,
Kuldeep.lal@icar.gov.in

## ABSTRACT

A new species of the genus *Pangasius,* is described based on 17 specimens collected from the Cauvery River, India. It can be distinguished from its sister species from South and Southeast Asia, by its widely placed, small and rounded vomerine and palatine tooth plates, longer maxillary and mandibular barbels, greater vertebrae count 50 (vs. 44–48), and smaller caudal peduncle depth (6.5–8.2% SL vs. 9.89–13.09% SL). The tooth plates of the new species closely resembles that of *Pangasius macronema* but can be clearly distinguished from the latter by having lesser gill rakers (16–19 vs. 36–45); a smaller eye (2.4–4.4% SL vs. 5.2–9.6% SL); and larger adipose-fin base (1.5–2.9% SL vs. 0.1–1.2% SL). The mitochondrial cytochrome c oxidase (COI) gene sequence of the new species shows the genetic divergence of 3.5% and 5.1% from *P. pangasius* and *P. silasi* respectively, the two sister species found in South Asia and India. The species delimitation approaches, Poisson Tree Processes (PTP) and assemble species by automatic partitioning (ASAP) clearly resolved that the *P. icaria* is distinct from its sister species. Phylogenetic position of the species with its sister species was evaluated using maximum likelihood and Bayesian analysis. The discovery of this previously unknown species of genus *Pangasius* from the Cauvery River of peninsular India indicates important biogeographical insight that this genus migrated till the southern division of Western Ghats.

## INTRODUCTION

The past few decades have witnessed the discovery of several new catfish species from southeast and south Asia (*Pouyaud, Teugels & Legendre, 1999*; *Roberts, 1999*; *Pouyaud & Teugels, 2000*; *Pouyaud, Gustiano & Teugels, 2002*; *Gustiano, Teugels & Pouyaud, 2003*) including some from the peninsular India (*Radhakrishnan, Sureshkumar & Ng, 2010*; *Babu, 2012*; *Ng, 2013*; *Lal et al., 2017*; *Dwivedi et al., 2017*). Pangasiid catfishes (*Pangasius* Valenciennes, 1840) are popular for their commercial value in both aquaculture and wild capture fisheries (*Roberts & Vidthayanon, 1991*) and are considered to be a delicacy. A total of 22 valid species are distributed in the river basins of southeast and south Asia (*Dwivedi et al., 2017*; *Fricke, Eschmeyer & Fong, 2021*). These have also been introduced in many countries for aquaculture. Among the valid species, only two species have been known from the Indian rivers, the most common is *Pangasius pangasius* (Hamilton, 1822), distributed in the rivers Ganges, Brahmaputra and Godavari, and the other, *Pangasius silasi* (*Dwivedi et al., 2017*) is described from the Nagarjuna Sagar Dam, in the Krishna River.

Previously, *Hora (1951)* had mentioned about the presence of *Pangasius* in the Stanley Reservoir (also known as Mettur Dam), which is one of the largest fishing reservoirs in South India. However, inadequate data led to consideration of the species as *Pangasius pangasius* only, though this raised hitherto unresolved query for the biogeographers of the time (*Silas, 1952b*). They assumed the absence of divergence in *Pangasius* genus, which probably migrated into peninsular India from Himalayan rivers. This is in contrast to some genera which are common between rivers of the Himalaya and the Indian peninsula, however, are represented by different species such as Silonia (family Schilbidae).

The explorations were conducted in the Cauvery River basin, Tamil Nadu, India and specimens of *Pangasius* were collected. In the present study, we used mitochondrial gene cytochrome c oxidase (COI) to support the morphological analysis. Morphometrics and molecular marker dataset analysis were used for deciphering phylogenetic relationships and species delimitation. Results explicitly indicated a distinct clade of the specimens from the Cauvery River, which is well differentiated from its congeners and described here as *Pangasius icaria,* a new species.

## MATERIALS & METHODS

Specimens described in present study, were collected from the commercial catches from landings along the locations, Mettur Dam, (Tamil Nadu). and upstream of Shivanasamudra falls, Chamarajanagar (Karnataka), on the Cauvery River basin in India. The details of samples and collection localities are presented in Table S1. The right-side pectoral fin was excised and preserved in 95% ethanol for molecular studies form individual specimen. Specimens were tagged with unique code, preserved in formalin (10%) and transported to the laboratory for examination and permanent preservation.

Data for meristic and morphometric measurements were generated following *Ng & Dodson (1999)*. Measurements were taken using a digital caliper (Mitutoyo Digimatic Caliper, Japan) to the nearest 0.1 mm. The body and head subunits were presented as percent (%) of standard length (SL). A total of 31 morphological characters were recorded

for comparative assessment with other pangasius species (*Roberts & Vidthayanon, 1991*; *Dwivedi et al., 2017*). Individual digital X-ray radiographs were used for vertebrae count following *Ng & Kottelat (2013)*. Morphometric measurements in proportion of standard length (SL) and head length (HL) of the holotype, followed by paratype were documented. The holotype and paratype accessions, examined in the study, were deposited in the notified repository (http://nbaindia.org/uploaded/pdf/notification/1%20designated%20repositories. pdf), National Repository and Fish Museum at ICAR-NBFGR, Lucknow, India.

## Explorations and permissions for field collections

In India Biological Diversity Act 2002 regulates access the bioresources and biosurveys (http://nbaindia.org/uploaded/act/BDACT_ENG.pdf). 1. Under Chapter II, section 2, it does not restrict public funded institutes for explorations and accessing bioresources for research purpose. Further, in explanation to section 4, it permits use of sharing research data for publication purpose. Therefore, NBFGR is a public funded research institute and is exempted from seeking specific permission under BDA 2002. Further, under Chapter IX, section 39(1) the designation of repositories are mentioned and NBFGR itself is such designated repository under the law (http://nbaindia.org/uploaded/pdf/notification/1% 20designated%20repositories.pdf) Further section 39(3) refers of depositing new taxon in such designated repositories and that is complied with in this work.

## Gene amplification and sequencing

Genomic DNA was extracted from the fin (stored in 95% ethanol) following the salting-out method (*Sambrook & Russel, 2001*). The partial mitochondrial cytochrome c oxidase subunit 1 (COI) gene was amplified using universal primers, forward primer Fish F1 (5′TCAACCAACCACAAAGACATTGGCAC3′) and reverse primer Fish R1 (5′TAGACTTCTGGGTGGCCAAAGAATCA3′) (*Ward et al., 2005*). The amplification was performed in 25 µL reaction which consisted of 1X reaction buffer (10 mM Tris, 50 mM KCl, 0.01% gelatin, pH 9.0), 1.5 mM MgCl$_2$, 200 µM of each dNTP, 5 pmol of each (F1 and R1) primer, 3U Taq polymerase (Genei, India) and approximately 50 ng genomic DNA. Thermal cycling was performed with an initial denaturation at 95 °C for 4 min, 35 cycles of 94 °C for 30 s., annealing at 52 °C for 40 s., extension at 72 °C for 45 s and final extension at 72 °C for 10 min. The amplicons were visualized on agarose gel and purified for bidirectional sequencing on ABI automated sequencer (Applied Biosystems, Waltham, MA, USA).

## Genetic analysis

The generated sequences were aligned using the ClustalW algorithm in BioEdit version 5.0.9 (*Hall, 1999*) along with other gene sequences of the genus *Pangasius* downloaded from NCBI GenBank. The sequences used for analyses are provided in Table S2. The estimates for inter-species genetic divergence was obtained using the Kimura 2P model (*Kimura, 1980*) implemented in MEGA-X (*Kumar et al., 2018*).

Maximum likelihood phylogenetic analysis was performed (500 bootstraps) using the software MEGA-X (*Kumar et al., 2018*). The outgroups used for rooting the tree included the COI sequences of *Horabagrus brachysoma* (Family: Horabagridae) and *Clupisoma*

*garua* (Family: Schilbeidae) both belonging to order *Siluriformes.* The best fit model and the partition scheme, for the present dataset was obtained from PartitionFinder2 (*Lanfear et al., 2017*), which revealed HKY+G to be the optimum for phylogenetic analysis. Data partitioning was done for all three codon positions, of which the first position was optimum. Bayesian analysis was performed using MrBayes v.3.1.2 (*Huelsenbeck & Ronquist, 2001*) with Markov Chain Markov Chain Monte Carlo sampling with four chains. A total of one million MCMC runs were executed and the trees and parameters sampled at every 1000 generations. The analysis was run until the deviation from split frequencies remained 0.01. The potential scale reduction factor was closer to one. A total of 25% of the samples were discarded as burn-in. The convergence of runs was observed in MCMC trace analysis package v1.7.2 (*Drummond & Rambaut, 2007*), while the tree topology was visualized in Figtree v1.4.4 (*FigTree, 2018*).

## Species delimitation

Recent studies have suggested that fine resolution can be obtained through employing multiple approaches for species delimitation (https://doi.org/10.1007/s12526-015-0319-7). In this study, we used two methods, assemble species by automatic partitioning (ASAP) (*Puillandre et al., 2012*) and the tree-based Poisson tree processes (*Zhang et al., 2013*). The ASAP calculates genetic distances between DNA sequences and designates partitions of species on the basis of the best ASAP score (*Puillandre, Brouillet & Achaz, 2021*). The overall analysis (this study and downloaded sequences) was used for partitioning the groups. The PTP delimits the species on a rooted phylogenetic tree on the basis of number of substitutions. Here, the PTP (http://species.h-its.org/ptp/) was implemented using optimized maximum likelihood tree constructed for COI gene using MEGA-X. The codon-partitioned COI dataset was used for analysis.

## Comparative material examined

*Pangasius pangaisus*: NBFGR/ PP 37 & 40, 2 ex., Jobra Barrage, the Mahanadi River, Odisha, October 2005.

*Pangasius pangaisus:* NBFGR/ PP 66–68, 3 ex., Pranhita River, tributary of river Godavari, Telangana, November 2012.

*Pangasius silasi*: NBFGR/72–78, 7 ex., 247.8–407.4 mm SL, Nagarjuna Sagar Dam, Telangana, India

*Pangasianodon hypothalamus*: NBFGR/PANPHYP.1–2, 450.5–489.4 mm SL, cultured pond, Mettur Dam, Tamil Nadu, India, July 2021.

*The electronic version of this article in Portable Document Format (PDF) will represent a published work according to the International Commission on Zoological Nomenclature (ICZN), and hence the new names contained in the electronic version are effectively published under that Code from the electronic edition alone. This published work and the nomenclatural acts it contains have been registered in ZooBank, the online registration system for the ICZN. The ZooBank LSIDs (Life Science Identifiers) can be resolved and the associated information viewed through any standard web browser by appending the LSID to the prefix* http:

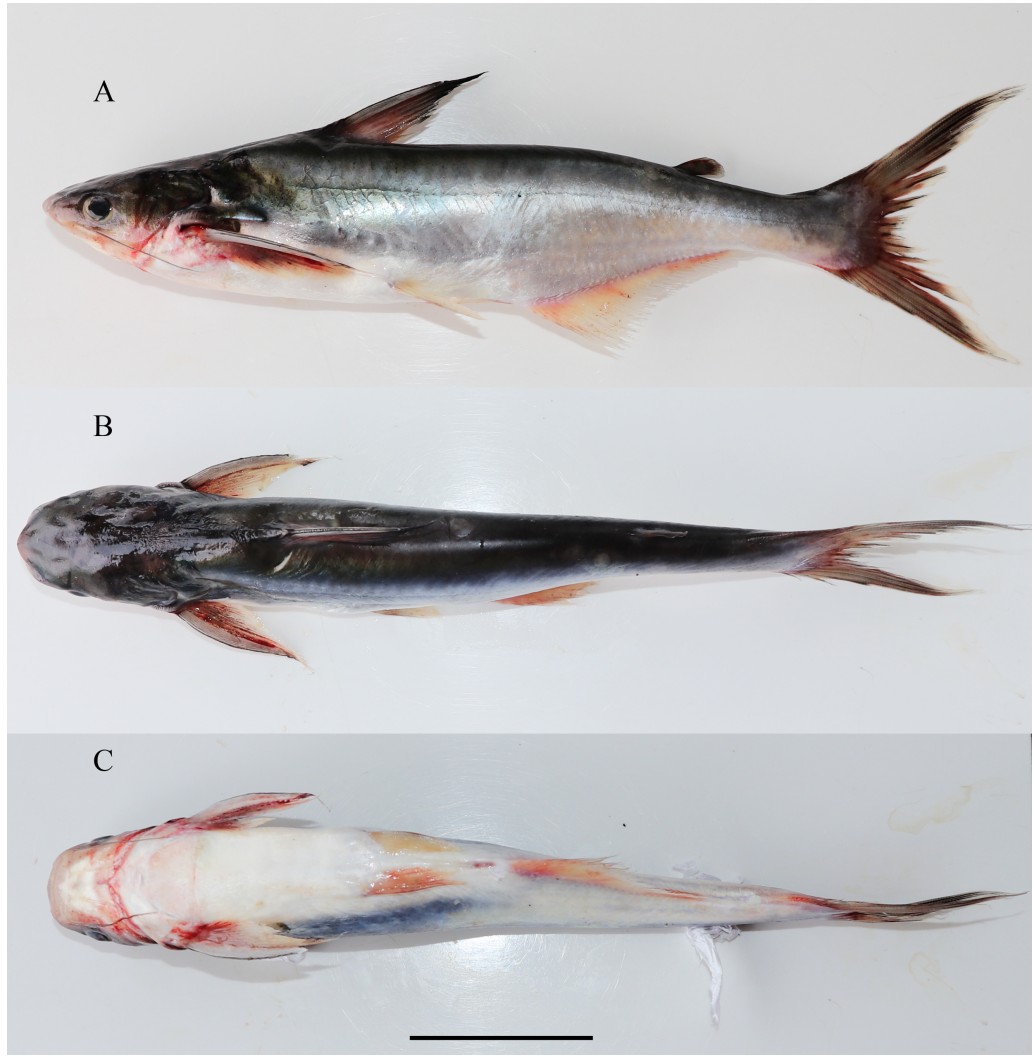

**Figure 1** **Lateral dorsal and ventral view of *Pangasius icaria*, holotype, NBFGR/ PANPTAM, 211.6 mm SL.** (A) Lateral; (B) dorsal; (C) ventral view of *Pangasius icaria*, holotype, NBFGR/ PANPTAM, 211. 6 mm SL, Tamil Nadu, Salem district, Mettur Dam, Cauvery River.

*//zoobank.org/. The LSID for this publication is:* **urn:lsid:zoobank.org:pub:C5FA8AA4-A223-4FB4-AF4F-D6399F9F3E02**. *The online version of this work is archived and available from the following digital repositories: PeerJ, PubMed Central SCIE and CLOCKSS.*

## RESULTS

*Pangasius icaria,* sp. nov.

Figs. 1–8; Table 1 and Table S4

**Holotype**. NBFGR/PANPTAM, 211.6 mm SL, India, Tamil Nadu, Salem district, Mettur Dam village, Mettur Dam, 11°48′47.1′N; 77°48′08.2′E, 30 June 2021, Kathirvelpandian and Kodeeswaran.

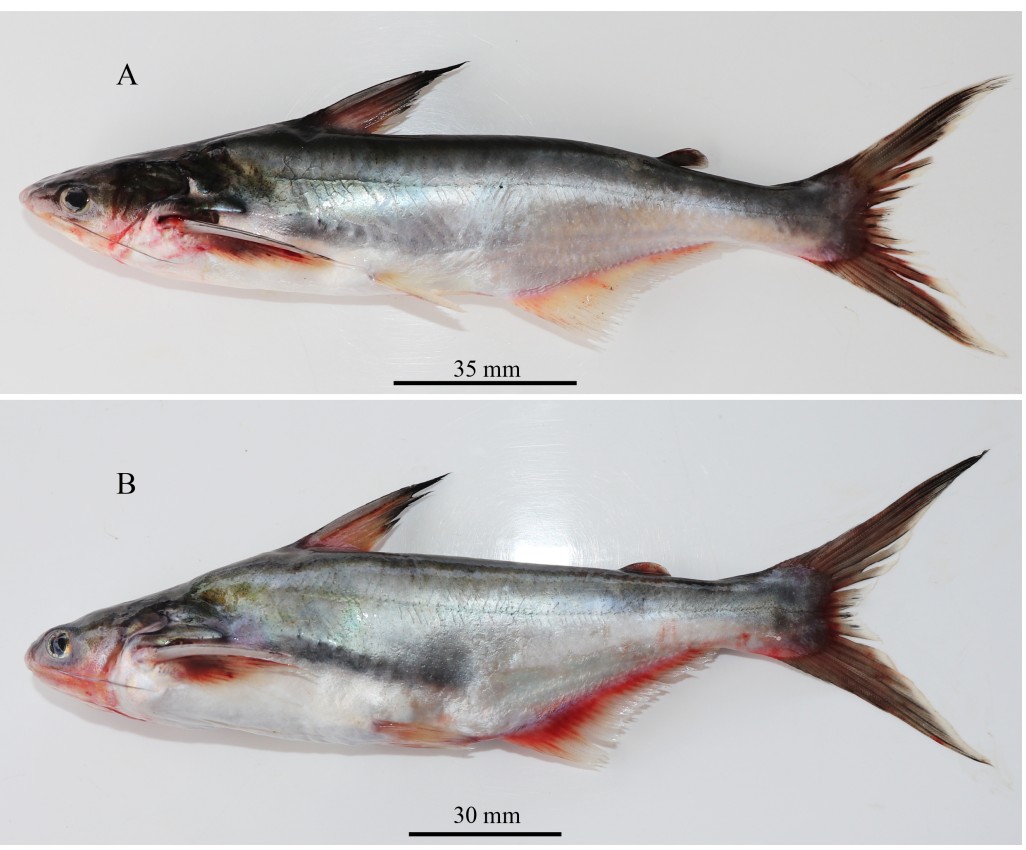

**Figure 2** Fresh coloration images (A) *Pangaisus icaria*, holotype, NBFGR/PANPTAM, 211.6 mm SL; (B) paratype, NBFGR/PANPTAM.1, 250.2 mm SL; (C) *Pangasiodon hypothalamus*, NBFGR/PANPHYP.1, 489.4 mm SL.

**Paratypes**. NBFGR/PANPTAM.1–11, 11 ex., 130.6–257.6 mm SL, same data as holotype. NBFGR/PANPTAM.12–15, 4 ex., 313–420 mm SL, upstream of Shivanasamudra falls, Cauvery river basin, Chamarajanagar, 12°16′11.6′N; 77°10′08.0′E, Karnataka, Charan, Basheer and Rahul.

**Diagnosis**. *Pangasius icaria* differs from all the sister species of south Asia by by the following combination of characters: a set of widely placed, small and rounded vomerine and palatine tooth plates, moderately rounded snout on dorsally viewed, maxillary barbel reaching beyond the base of the pectoral spine, eye diameter 2.4–4.4% SL, smaller-interorbital distance 10.1–12.4% SL, caudal peduncle depth 6.5–8.2% SL, filamentous first dorsal- and pectoral-fin ray, gill rakers 16–19, 50 vertebrae and a reddish dorsal-, anal- and pectoral-fin base.

**Description.** Morphometric data of the *P. icaria* as in Table 1. Head depressed and snout rounded. Mouth sub-terminal, upper jaw tooth band faintly exposed when mouth closed (Figs. 1 and 2). Relatively shorter median longitudinal groove, reaching the base of occipital process, occipital process terminates before the dorsal-spine plate (Fig. 3). Lower jaw slightly curved or rounded. Mouth moderately wide vomerine and palatine

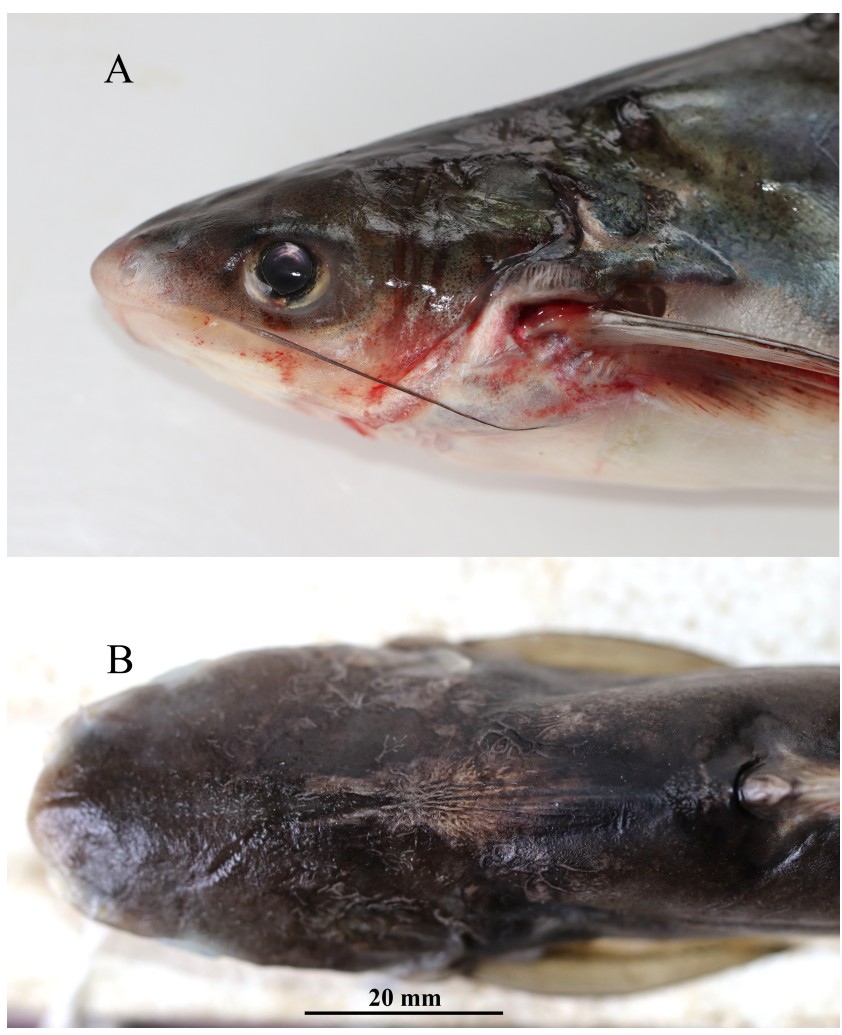

**Figure 3** *Pangaisus icaria*, holotype, *Pangaisus icaria*, holotype, NBFGR/ PANPTAM, 211. 6 mm SL; (A) lateral view showing the maxillary barbel extension up to the pectoral-fin base; (B) dorsal view showing median groove of holotype.

tooth bands are small and rounded and widely placed from each other (Fig. 4). Barbels well developed, maxillary barbels long extending beyond the pectoral-spine base; mandibular barbels long extending beyond gill opening. All the fin-rays branched, dorsal-fin rays seven, originating vertical through half the length of the pectoral spine. Dorsal spine long, sturdy with blunt tip, ventral margin serrated dorsal margin weakly serrated. Adipose fin short, its base 2.2% SL. Pectoral-fin rays 11, with filamentous extension reaching the base of the pelvic fin; pelvic-fin rays 6, its origin vertical through the posterior end of the dorsal-fin base, not reaching anal fin; anal fin with 27–29 rays, terminates well-before caudal-fin base. Dorsal procurrent caudal-fin rays 16–17; ventral procurrent caudal-fin rays 13–15; principal caudal-fin rays 17 (8+9), deeply forked.

**Coloration.** Live specimens: Dorsal surface bronze green to dark-green, ventral silvery white; dorsal-fin base reddish to bright-orange with black margin; caudal fin reddish-black

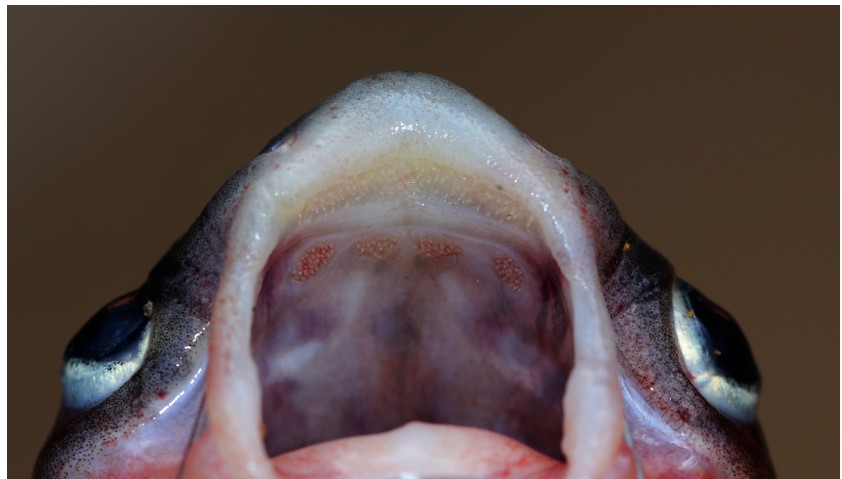

**Figure 4** Maxillary and palatal dentition of *Pangasius icaria* holotype, NBFGR/ PANPTAM, 211. 6 mm SL.

with creamy white margin; pectoral fin dark-red with white margin; pelvic fin creamy-red white; anal-fin base brick-red becoming translucently white towards the margin; adipose fin dark-grey base with white margin (Figs. 2A, 2B). Eye pupil black with golden margin. Maxillary barbel dark-greyish in colour, mandibular barbel creamy-white. In preserved condition; dorsal surface dark, ventral region creamy-white or pale-grey; all the fins beige to tan with pale yellowish base.

**Feeding.** Gut content analysis of three samples revealed the presence of the molluscs, *Bellamya bengalensis* (Fig. 5A); X-ray radiographs also revealed the presence of numerous molluscs in the gut (Fig. 5B).

**Habitat and distribution**. Presently known only from Cauvery River basin collected from two locations, at Mettur Dam, and in the upstream of Shivanasamudra Falls, Chamarajanagar, Karnataka. The species was collected using a gill net at a depth of 5–15 m during the discharge of water from the main dam (Fig. 6).

**Etymology**. The species is named after the Indian Council of Agricultural Research (ICAR) and used its abbreviated form. ICAR is the parent organization for NBFGR, which has conducted this research.

**Comparison.** *Pangasius icaria* differs from all other sister species by having widely placed, small and rounded vomerine and palatine tooth plates (*vs.* single curved and joined palatine and vomerine tooth plates in *Pangaisus larnaudii* Bocourt 1866; uninterrupted broad, strongly curved, palatal tooth band in *Pangasius krempfi* Fang & Chaux 1949 and *Pangasius sanitwongsei* Smith 1931; squared vomerine tooth plate with smaller palatine tooth bands in *Pangasius myanmar* (*Roberts & Vidthayanon, 1991*) and *Pangasius polyuranodon* Bleeker 1852; wider vomerine tooth plate with large palatine plates in *Pangasius bocourti* Sauvage 1880 and *Pangasius djambal* Bleeker 1846; single large median palatal tooth plate in *Pangasius humeralis* Roberts 1989, *Pangasius kinabatanganensis* (*Roberts & Vidthayanon, 1991*), *Pangasius lithostoma* Roberts 1989 and *Pangasius nieuwenhuisii* (Popta 1904).

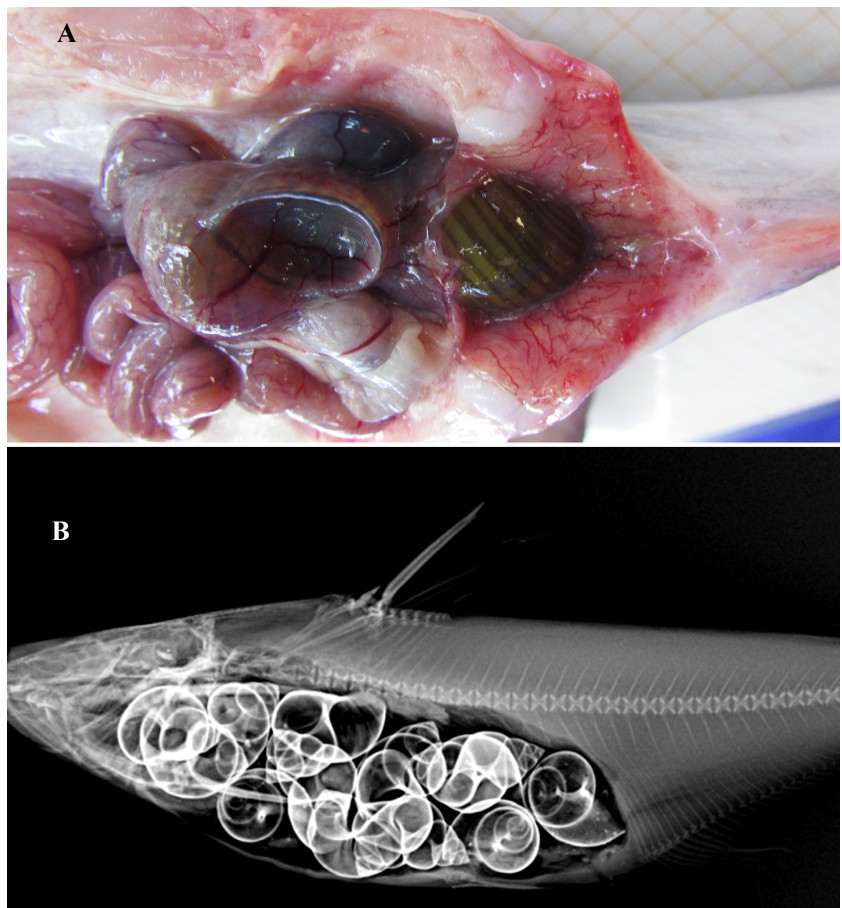

**Figure 5** **Gut contents of *Pangasius icaria*.** (A) Gut of the *Pangasius icaria* with *Bellamya bengalensis*; (B) X-ray radiographs showing the gut with molluscs.

Further, *P. icaria* differs from sister species from south Asia, *P. pangasius,* by having more vertebral count 50 (*vs.* 44 in *P. pangasius*); lesser gill rakers 16–19 (*vs.* 23); four small and round tooth plates which were widely separated from each (*vs.* elongated and curved toothplates placed close to each other); lesser caudal peduncle depth 6.5–8.2% SL (*vs.* 9.8–13.1% SL); smaller-interorbital distance 10.1–12.4% SL (*vs.* 13.8–14.7% SL); larger maxillary barbel length, 11.1–23.9% SL (*vs.* 10.56–10.87% SL). *Pangasius icaria* differs from its sister species *P. silasi* by having a widely separated and rounded small tooth plates (*vs.* uninterrupted combined tooth plates in *P. silasi*); more vertebral counts 50 (*vs.* 48); lesser gill rakers 16–19 (*vs.* 21); longer pectoral- and dorsal-spine length, 15.1–19.7% SL; 14.4–18.6% SL (*vs.* 4.02–9.42% SL; 8.46–11.62% SL); longer caudal peduncle length, 14.6–19.2% SL (*vs.* 11.04–14.88% SL); lesser caudal peduncle depth 6.5–8.2% SL (*vs.* 10.40–12.24% SL); shorter adipose-fin base length, 1.5–2.9% SL (*vs.* 3.96–5.87% SL); larger maxillary barbel length, reaching pectoral-fin base, 11.1–23.9% SL (*vs.* relatively shorter in length, 10.94–13.38% SL); smaller-interorbital distance 10.1–12.4% SL (*vs.* 12.66–15.11% SL). *Pangasius icaria shares* a similar vertebral count and have a filamentous extension on

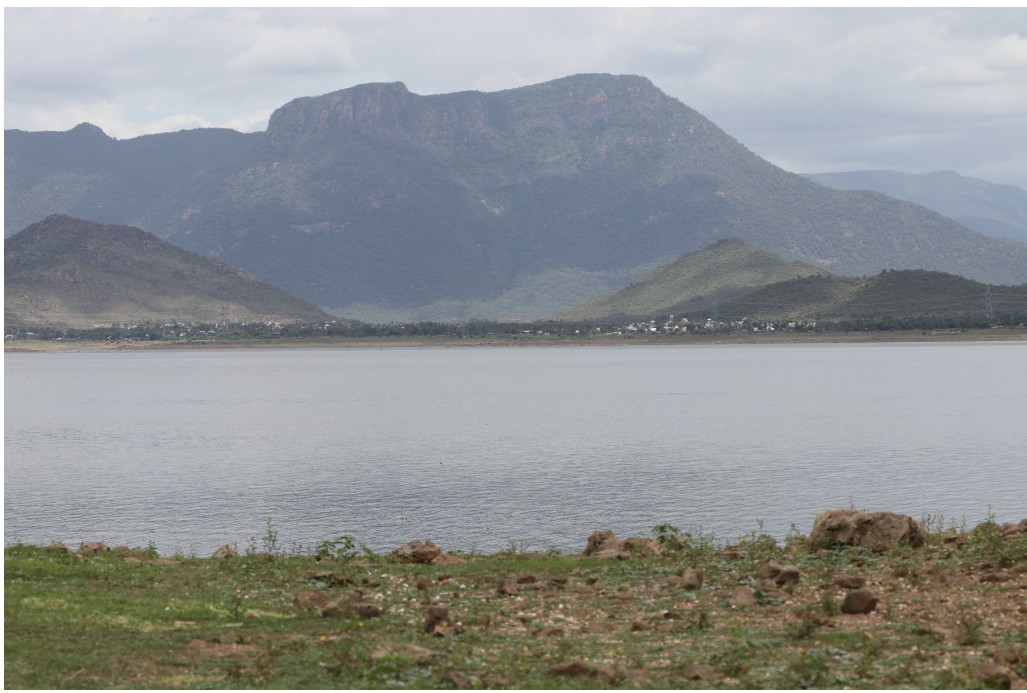

**Figure 6** Habitat of *Pangasius icaria* in the river Cauvery, Tamil Nadu (11°54′24.4″N; 77°46′32.4″E). Photo Source Credit: ICAR-NBFGR.

the dorsal and pectoral fin with *Pangasius santiwongsei* Smith, 1931 but readily differs by having a smaller mouth width gape 6.2–9.0% SL (*vs.* 17% SL). Further, the new species shares a similar vomerine palatine tooth band with *Pangasius macronema* Bleeker, 1850 but readily differs by having more vertebrae 50 (*vs.* 42–45 in Thailand species and 41–43 in Borneo species); gill rakers 16–19 (*vs.* 36–45); having smaller eye, 2.4–4.4% SL (*vs.* 5.2–9.6% SL); larger adipose-fin base, 1.5–2.9% SL (*vs.* 0.1–1.2% SL).

## Molecular analysis

A total of 11 individual COI sequences (575 bp) of *P. icaria* were characterised by two haplotypes, displaying 574 variable sites and one singleton. Sequence analysis revealed average nucleotide frequencies as T = 29.9%, C = 26.8%, A = 25.4%, G = 17.9%. The mean nucleotide diversity was found to be 0.0003+/−0.0004. The gene sequences have been submitted to NCBI with accession no. OK480013–OK480014, OK480046–OK480054. Substantial genetic divergence (2.96%) in *Pangasius icaria* from its congener *P. silasi*. Further, the new species differs from other pangasid species from south Asia *P. pangasius* and southeast Asia *Pangasius mekongensis* (*Gustiano, GG & Pouyaud, 2003*), *Pangasius conchophilus* (*Roberts & Vidthayanon, 1991*, *Pangasius bocourti* Sauvage, 1880), *P. sanitwongsei, Pangasius krempfi* (Fang & Chaux, 1949) with the genetic distances of 4.74%, 7.2%, 8.1%, 9.0%, 10.0% and 11.1% respectively. These results supported that the pangasius specimens studied from river Cauvery, belong to a new and previously undescribed species. Phylogenetic analysis using MEGA X and Bayesian inference analysis

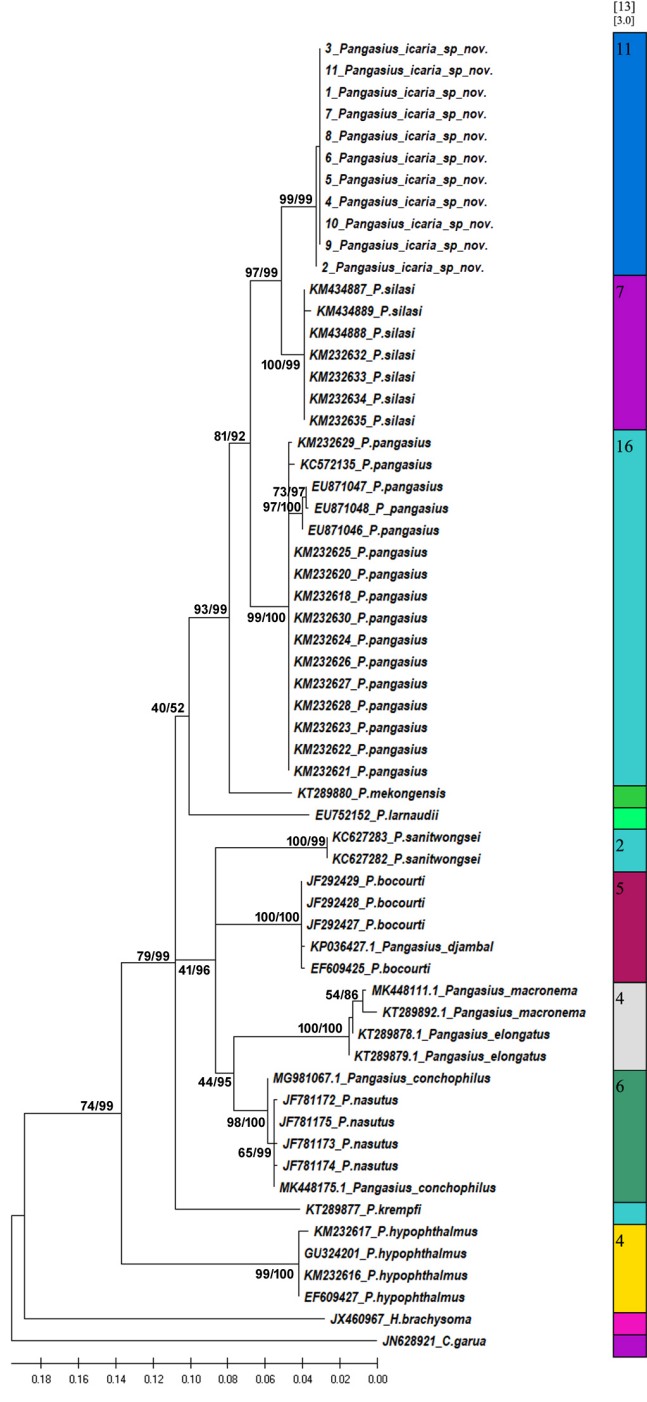

**Figure 7** **Bayesian analysis of *Pangasius* species.** The evolutionary status of *Pangasius icaria* in relation with other species of genus *Pangasius* using Bayesian analysis.

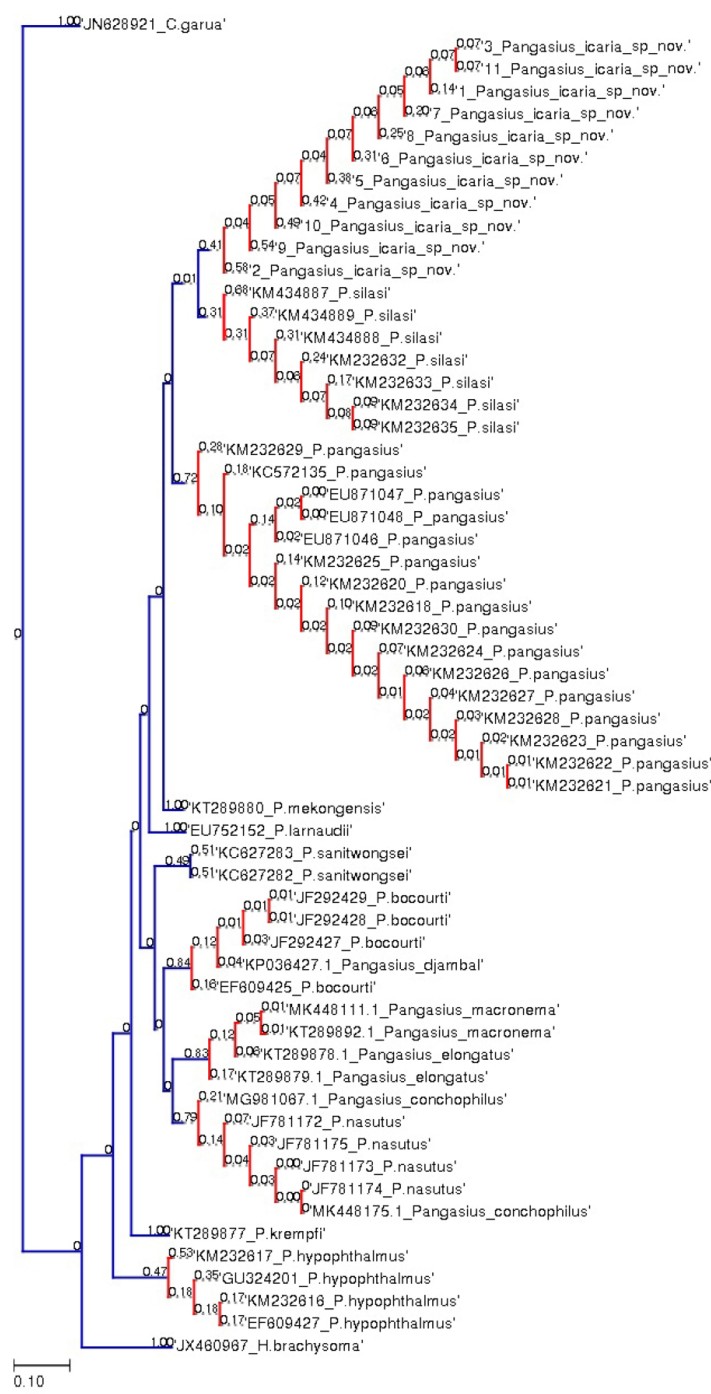

**Figure 8  Species delimitation of *Pangasius* species.** Phylogenetic reconstruction of *Pangasius* species with outgroups *H. brachyosoma* and *C. garua*. Node values are presented as ML/BI. Right: Species delimitation using distance-based ASAP scores (ABGD).

**Table 1  Morphometric data of *Pangasius icaria* compared with *Pangasius silasi* and *Pangasius pangasius*.**

| | *Pangasius icaria* sp. nov. (Holotype) | *Pangasius icaria* sp. nov. Mean (Range) $N = 16$ | SD | *P. silasi* (H) | *P. silasi* (P) | *P. pangasius* |
|---|---|---|---|---|---|---|
| Standard length (mm) | 211.6 | 130.6–420.0 | | | | |
| % Standard length | | | | | | |
| Head length | 19.8 | 20.3 (19.3–21.4) | 0.6 | 19.77 | 17.53–21.01 | 14.03–25.94 |
| Head width | 15.1 | 14.9 (13.9–15.9) | 0.6 | 14.91 | 14.22–16.51 | 14.38–16.26 |
| Head depth | 15.1 | 15.0 (13.0–17.3) | 1.2 | 14.41 | 13.37–16.43 | 10.86–18.21 |
| Predorsal length | 32.9 | 34.2 (33.0–35.6) | 0.8 | 36.62 | 33.42–36.08 | 31.69–43.29 |
| Preanal length | 54.2 | 56.8 (50.9–60.9) | 2.5 | 57.72 | 59.05–62.76 | 53.26–65.63 |
| Prepelvic length | 41.9 | 43.6 (41.7–45.7) | 1.2 | 42.96 | 43.83–47.53 | 41.38–51.15 |
| Prepectoral length | 19.6 | 19.6 (18.2–21.0) | 1.0 | 17.11 | 15.37–24.00 | 16.36–26.47 |
| Body depth at anus | 17.5 | 19.7 (16.1–23.5) | 2.2 | 19.87 | 19.56–26.65 | 20.76–29.05 |
| Length of caudal peduncle | 18.1 | 16.9 (14.6–19.2) | 1.3 | 12.40 | 11.04–14.88 | 9.21–15.83 |
| Depth of caudal peduncle | 6.8 | 7.3 (6.5–8.2) | 0.5 | 11.27 | 10.40–12.24 | 9.89–13.09 |
| Pectoral spine length | 18.4 | 17.7 (15.1–19.7) | 1.5 | 6.39 | 4.02–9.42 | 9.46–19.59 |
| Pectoral fin length | 23.2 | 19.0 (12.6–22.5) | 2.3 | 19.00 | 13.39–17.46 | 11.38–21.32 |
| Dorsal-spine length | 16.1 | 15.9 (14.4–18.6) | 1.2 | 7.93 | 8.46–11.62 | 15.72–21.91 |
| Length of dorsal fin base | 8.4 | 7.6 (6.7–9.4) | 0.8 | 10.62 | 6.59–10.02 | 6.35–13.58 |
| Pelvic fin length | 12.2 | 11.9 (10.7–13.0) | 0.7 | 9.56 | 7.95–12.01 | 10.06–14.49 |
| Length of anal fin base | 25.9 | 25.7 (23.8–27.9) | 1.2 | 28.37 | 22.46–28.65 | 27.03–31.43 |
| Caudal fin length | 25.5 | 24.7 (22.9–28.9) | 2.0 | 26.86 | 22.68–32.46 | 18.51–22.86 |
| Length of adipose fin base | 2.0 | 2.2 (1.5–2.9) | 0.5 | 4.88 | 3.96–5.87 | 2.10–6.90 |
| Maximum height of adipose fin | 2.3 | 2.1 (1.8–3.0) | 0.6 | 1.42 | 1.35–2.45 | 1.55–2.77 |
| Dorsal fin to adipose distance | 33.9 | 33.0 (30.5–36.0) | 1.6 | 32.56 | 33.85–35.87 | 17.50–40.96 |
| Post adipose distance | 24.5 | 23.6 (19.1–43.2) | 5.4 | 82.37 | 80.69–82.92 | 79.01–87.85 |
| Snout length | 7.7 | 7.8 (7.3–9.2) | 0.5 | 4.62 | 3.54–8.06 | 5.74–6.14 |
| Mouth width | 8.4 | 8.8 (7.7–10.3) | 0.6 | 8.37 | 8.37–8.66 | 8.23–8.84 |
| Mouth width gape | 6.7 | 7.2 (6.2–9.0) | 0.7 | 7.87 | 7.40–7.87 | 6.80–7.63 |
| Inter-orbital distance | 11.1 | 11.4 (10.1–12.4) | 0.7 | 12.64 | 12.66–15.11 | 13.8–14.72 |
| Eye diameter | 3.6 | 3.7 (2.4–4.4) | 0.7 | 3.13 | 2.83–4.22 | 2.28–2.60 |
| Maxillary barbel length | 14.8 | 15.4 (11.1–23.9) | 3.4 | 10.94 | 10.94–13.38 | 10.56–10.87 |
| Mandibular barbel length | 10.1 | 10.3 (5.3–14.4) | 2.3 | 6.07 | 6.07–9.16 | 5.73–7.79 |
| Premaxillary tooth plate transverse width | 5.6 | 5.6 (4.5–7.1) | 0.6 | 6.38 | 6.38–7.13 | 6.75–7.56 |
| Palatal tooth plate transverse width | 3.9 | 4.2 (3.3–4.9) | 0.4 | 4.68 | 4.68–5.43 | 5.34–5.92 |

clearly depicted that all *P. icaria* samples clustered together and were distinct from the closest species *P. silasi*. (Fig. 7). Inter and intra species genetic distances are illustrated in Tables S3A & S3B.

The species delimitation (PTP) analysis (this study and downloaded sequences) resulted in delimitation of 11 groups (Fig. 8) excluding the outgroup species. *P. icaria* was delimited clearly from others. The findings of distance-based ASAP were also concordant to PTP. The graphical pattern clearly differentiated the samples of *P. icaria*, with *P. silasi* and other

Pangasiid species (Figs. 7 & 8). The results are in concurrence with morpho-meristic measurements.

## DISCUSSION

In South Asia, the genus *Pangasius* is represented by two known species, to date. *P. pangasius* has a wide distribution in the rivers of Ganga, the Brahmaputra in India, Bangladesh and the Irrawaddy basin in Myanmar. The results from species delimitation approaches indicate consensus support for the presence of groups within genus Pangasius and provide clear and distinct monophyly of *P. icaria*. The new species reported here from a peninsular Cauvery River is clearly distinct from *P. silasi,* which appears to be its nearest neighbour. The species, *P. silasi* which is endemic to peninsular India and known only from the Krishna River system, so far (*Dwivedi et al., 2017*). It is interesting to note that feeding habit of *P. icaria* is similar to *P. silasi* (*Ajith Kumar, Santosh & Lal, 2020*) and both are molluscivorous.

The discovery of *P. icaria* increases the number of Pangasiid catfishes in south Asia to three. *Pangasius* species can be easily distinguished by the dorsal view of the head and vomerine, palatine tooth plates as suggested by *Roberts & Vidthayanon (1991)*. The new species is easily distinguishable from its known sister species, in south Asia, *P. pangasius,* and *P. silasi* by possessing a small, round and widely separated pair of tooth plates and filamentous fin extension. A common and widely cultivated Pangasiid catfish *Pangasianodon hypothalamus* (Sauvage, 1878) in India was also collected from the cultured pond of the same locality of the new species. The new species differs from the latter by having longer maxillary and mandibular barbels (*vs.* very short in *P. hypothalamus*). Genetic analysis also reveals the valid species status of the new species described in this study. Locally, the species is well known among the fisherfolks as 'Aie Keluthi' in the vernacular Tamil language. It is likely that *Talwar & Jhingran (1991)* have documented this new described species as *P. pangasius* from the Cauvery River, bearing the same vernacular name of 'Aie Keluthi'. Similarly, *Silas (1952a)* also documented *P. pangasius* from the Cauvery river system which might be the species described here. In the study of the evolutionary status of fishes of Himalayan and peninsular rivers, *Silas (1952b)* documented that fishes found that peninsular rivers of India harbour the genera common to Himalayan rivers but are represented by the alternate species. This was attributed to the probable migration of Malayan elements, during the Pleistocene period from Assam Himalayas to peninsular India (*Hora, 1951*; *Menon, 1980*) and named as Satpura hypothesis by *Hora (1949)*. The geological changes and vicariant alteration of river courses causing separation of migrating populations consequently resulted in the observed speciation in the peninsular rivers. The separation of these species might have occurred during middle Pleistocene (around 0.78–0.126 Mya) and led to the evolution of endemic fish species distributed in the peninsular rivers (*Briggs, 2003*). The present results did not reveal presence of any specimen which is similar to *P. pangasius,* the most common species found in India. *Dwivedi et al. (2017)* validated the presence of *P. pangasius* only upto the rivers Mahanadi and Godavari (with minor morphological difference), however, discovered an undescribed species in the Krishna River, named as *P.silasi.* Therfore, the earlier reports of

*P. pangasius* in the Krishna River (*Ramakrishnaya, 1986*) were not confirmed by the later studies (*Dwivedi et al., 2017*; *Ajith Kumar, Santosh & Lal, 2020*). The Mahanadi, Godavari, Krishna and Cauvery rivers are the large independent basins in peninsular India, which flow eastward to drain into Bay of Bengal. The Krishna River has the Mahanadi and Godavari rivers on its northern side and the Cauvery River on its south. The discovery of this new species, *P. icaria* from the Cauvery River confirms that the genus *Pangasius* has extended distribution upto the southern division of the Western Ghats and has undergone speciation, like other Himalayan species that have same genus but are represented by alternate species in the peninsular India (*Silas, 1952b*) for example *Silonia silondia* (Ganges and Mahanadi) were represented as *S. childreni* in peninsular rivers. This addresses the earlier paleobiogeographic conflict that the genus *Pangasius* might be a late migrant and has not undergone the species divergence (*Silas, 1952b*; *Menon, 1980*).

In the present study, the *P. icaria*, is described in collections from two different and distant locations in the Cauvery River. The rivers of peninsular India flows through the Western Ghat region, which is part of the designated mega biodiversity hotspots and is known for endemism. The future programs on fine-scale explorations of these rivers will be significant in delimiting the distribution ranges of the newly discovered pangasius species (*P.silasi and P. icaria*) for documentation of within species genetic diversity, their conservation strategies and also with the possibility of new undescribed species.

## CONCLUSIONS

A new species of the genus *Pangasius* is described that was collected from the Cauvery River, Tamil Nadu, India. This new discovery of *P. icaria* clearly highlights the native presence of genus *Pangasius* in peninsular India and is represented by two recorded divergent species. Future research and explorations are needed to ascertain the distributional range of this endemic species for devising conservation and management of the species and also to evaluate for its aquaculture utilization potential.

## ACKNOWLEDGEMENTS

The authors extend gratitude to the Director of the ICAR-National Bureau of Fish Genetic Resources for facilities and encouragements. The authors are also grateful to Tamil Nadu Govt. Fisheries Department, Mettur Dam for their help during the sample collections.

### Funding
The work was funded by ICAR-NBFGR under the program's Consortium Research Platform on Agrobiodiversity (FISHNBFGRCIP 201500700172) and ''Exploration and cataloguing of the fish diversity from Cauvery River basin'' (FISHNBFGRSIL201600400185). The funders had no role in study design, data collection and analysis, decision to publish, or preparation of the manuscript.

## Grant Disclosures

The following grant information was disclosed by the authors:

ICAR-NBFGR under the program's Consortium Research Platform on Agrobiodiversity: FISHNBFGRCIP 201500700172.

Exploration and cataloguing of the fish diversity from Cauvery River basin: FISHNBFGRSIL201600400185.

## Competing Interests

The authors declare there are no competing interests.

## Author Contributions

- Kathirvelpandian P.V. Ayyathurai performed the experiments, analyzed the data, authored or reviewed drafts of the article, and approved the final draft.
- Paramasivam Kodeeswaran performed the experiments, prepared figures and/or tables, and approved the final draft.
- Vindhya Mohindra analyzed the data, prepared figures and/or tables, authored or reviewed drafts of the article, and approved the final draft.
- Rajeev K. Singh analyzed the data, prepared figures and/or tables, authored or reviewed drafts of the article, and approved the final draft.
- Charan Ravi performed the experiments, prepared figures and/or tables, and approved the final draft.
- Rahul Kumar performed the experiments, prepared figures and/or tables, and approved the final draft.
- Basheer Saidmuhammed Valaparambil performed the experiments, authored or reviewed drafts of the article, facilitated collection of right specimens and accessioning, and approved the final draft.
- Ajith Kumar Thipramalai Thangappan conceived and designed the experiments, authored or reviewed drafts of the article, facilitated collection of right specimens and accessioning, and approved the final draft.
- Joykrushna Jena conceived and designed the experiments, authored or reviewed drafts of the article, and approved the final draft.
- Kuldeep K. Lal conceived and designed the experiments, performed the experiments, analyzed the data, authored or reviewed drafts of the article, and approved the final draft.

## Field Study Permissions

The following information was supplied relating to field study approvals (i.e., approving body and any reference numbers):

National Bureau of Fish Genetic Resources.

## DNA Deposition

The following information was supplied regarding the deposition of DNA sequences:

The cytochrome C oxidase I sequences are available at GenBank: OK480013–OK480014, OK480046–OK480054.

## Data Availability

The sequence information and raw morphological data are available in the Supplementary Files.

## New Species Registration

The following information was supplied regarding the registration of a newly described species:

Publication LSID:

urn:lsid:zoobank.org:pub:C5FA8AA4-A223-4FB4-AF4F-D6399F9F3E02

Pangasius icaria species LSID:

urn:lsid:zoobank.org:act:7713F0B9-92B8-45EC-BBF4-B1D662658291

Pangasiidae Lal family LSID: urn:lsid:zoobank.org:act:B63817A7-7596-43CE-8E9F-915140314DC7.

## Supplemental Information

Supplemental information for this article can be found online at http://dx.doi.org/10.7717/peerj.14258#supplemental-information.

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
