# Peer review of "Description of a new Pangasius (Valenciennes, 1840) species, from the Cauvery River extends distribution range of the genus up to South Western Ghats in peninsular India"

_PeerJ, doi:10.7717/peerj.14258_

## Round 0.1 · original submission · Major Revisions

It's a fantastic job to discover a new species, but requires a lot of accurate information and robust analyses. There's a need to satisfy all the reviewers' queries and suggestions before reaching the conclusion that a new species has been discovered.

Reviewer 1 ·

Excellent Review

This review has been rated excellent by staff (in the top 15% of reviews)
EDITOR COMMENT
Thanks to the reviewer for a very detailed and constructive review that can help the authors to improve the manuscript substantially to be accepted for publication.

Basic reporting

English: Authors need to seek help to improve their English grammar. For example, Line 125-126, "partitioned the groups clearly" should be "clearly partitioned the groups". Also authors should use technical language. For example, line 94, "PCR was done" should be "PCR was performed". Line 228-229, "nearest congeneric species", are technically called as "sister taxa" or "sister species". Line 247, "its nearest neighbor" should be "sister taxa" or "sister species".

Literature: Hora’s Satpura hypothesis should be attributed to the paper “Hora, S. L. (1949) Satpura Hypothesis of the distribution of the Malayan flora and fauna to peninsular India. Proceedings of the National Institute of Science India, 15, 309–314.” Satpura hypothesis has been refuted by several recent studies and authors do not refer to any of these studies. This is not correct. Authors should have presented both, arguments for and against the hypothesis. In general, the authors should have discussed more about peninsular Indian fish fauna and how molecular phylogeny is changing our understanding by citing relevant recent references.

Article structure: Tables and figures are not prepared properly. Table 1 show measurements as % standard length, although in methods authors mention that the measurements are expressed as % total length. Also, all measurements are expressed as % standard length and no measurements are expressed as % head length as is mentioned in the methods. As per the taxonomic methods in ichthyology, the subunits of head should be expressed as % head length. Figure 1 should have been against a darker background so make the features discrete. Figure 3, the photographs are out of focus and have bad lightening. Figure 4 is good, but authors should provide similar photographs of sister taxa or sister group to make the distinction in character clear. Figure 4 is unnecessary because there is no point in comparing species of two distinct genera. Figure 6 is redundant. At the most, authors can provide it as supplementary information. Figure 9 is illegible and authors should check the font size requirement for the journal. Also, figure 9 caption is incomplete. Authors do not mention which tree NJ or Bayesian is actually shown here. The outgroups are not mentioned. It is not mentioned why some nodes have no support values. Figure 10 is a result of not understanding how ASAP works. ASAP provides the best partition scheme based on the scores and authors need to show only the best partition in the data. There is no need to show other partitions. Figure 11 is based on erroneous analysis (explained elsewhere). Authors should not just show the central tendencies in the time estimates, they should also estimate the errors in estimation and show the confidence interval of time estimates. The calibration points also need to be indicated (if that is how the analysis is performed as per the methods).

Self-contained with relevant results to hypotheses: The study deals with three hypotheses (even though they are not explicitly stated) and authors make statements without providing proper evidence for the same. The first hypothesis is the new species. Authors, provide phylogeny based on NJ rather than ML and this is not acceptable because NJ is a distance based method and not a character based method so it does not account for homoplasy and does not look at the patterns in nucleotide substitutions. Reciprocal monophyly of the clades do not necessarily suggest presence of a distinct species as haplogroups can also show such monophyly. Authors should have provided proper evidence based on genetic species delimitation methods. Although ASAP is provided authors do not show the results properly, like what was the best partition as identified by ASAP and rather show all the partitions in Figure 10 which is redundant. Authors should have also performed PTP (based on both ML and Bayesian analysis), GMYC or BPP to support the hypothesis of molecular delimitation of species. Second hypothesis is the recent evolution of the species. The molecular dating performed by authors is erroneous so the stamen of recent evolution of the species is wrong. Authors should have used ML or Bayesian tree rather than NJ for time tree analysis because NJ does not correct errors due to homoplasy and does not look at the nucleotide substitution rates. SO NJ cannot provide a phylogeny that reflect the evolutionary rates. Also, authors mention in the methods that calibrations were used without providing these calibrations in the methods or in the figure, making the entire analysis doubtful. Authors also do not provide uncertainty in date estimates (like 95% HPD) to check the reliability of the estimates. Third hypothesis is the statement provided in lines 278-280 which first of all make no sense and there is no evidence in the manuscript that support it.

Experimental design

Originality: Although the manuscript might be within the aims and scope of the journal and the species novelty might be true, most of the analysis, especially regarding the biogeographical significance highlighted in the title of the manuscript, is erroneous and not properly supported.

Research questions: The research questions are not well defined in the introduction. Authors should define the objectives and hypothesis clearly in the introduction. The reverence of the study, with respect to the current studies on ichthyofauna and ichthyofaunal studies in the southern India, is also not properly stated in the introduction or discussed. In general, the importance of the study to fill the knowledge gaps in fauna of southern India and catfish taxonomy in general are not stated.

Investigation: The investigation is not rigorous. There are many methodological lacunas. For example, authors ignore species such as Pangasius macronema, P. conchophilus, P. elongatus and P. djambal in their genetic analysis, even though there are multiple COI sequences for these species in GenBank database. Genetic delimitation of species is not rigorous. Although authors have performed ASAP the best partition identified by ASAP is not shown (authors show all partition, which is redundant information). Further, authors should have performed other genetic species delimitations such as PTP, GMYC or BPP.

Methods: Methods are not described with sufficient details. For example, time tree analysis vaguely mentions one calibration constraint but that has not been provided. Further, since authors have not tested statistically, whether the strict clock can be applied for the data, using single calibration point (if at all any point is used) makes little sense. In addition, the model used for distances calculated for NJ is not provided so NJ tree cannot be reproduced. Primer sequences or at least the names of the primers are not provided.

Validity of the findings

Importance of the manuscript: The species might be novel. So describing it is definitely beneficial. Unfortunately, the study is not performed properly and the manuscript is poorly written so the species description is getting overshadowed. Authors should rewrite the manuscript to explain the hypothesis (regarding species and biogeography) clearly in the introduction and set the proper flow to the arguments. The discussion need to explain the results in larger context of species diversity in southern India and within catfishes in general.

Data availability: Authors provide data for the range of morphometric characters. They can provide morphometric data of all the specimens in supplementary information. Authors should also provide data on meristics of the species. Authors should provide a table for sequences used in the current study with their proper citation and GenBank accession numbers.

Conclusions: The paper is not concluded properly. Authors’ statement (287-288) that the new discovery clearly highlights that genus migrated southward in peninsular India and diverged into two species is not supported by any analysis. Authors have assumed Hora’s Satpura to be true and make a vague statement based on NJ tree rather than using ML tree, which accounts for homoplasy and nucleotide substitution rates. Even the time tree analysis is flawed. So the analysis provided in the manuscript does not support the conclusion.

Additional comments

Although the species described in the manuscript might be valid. This is a poorly conceived and written study. In the abstract, authors provide some diagnostic characters (lines 26-28) about caudal peduncle length, interorbital distance and adipose dorsal fin base length and all these characters are overlapping with the congeners. So using these characters for diagnosis is just wrong. Further, authors compare the genetic distance between two congeners (without stating why, for example because they are sister taxa) while neglecting the remaining 20 valid species (or at least those species for which genetic data are available). The last statement in abstract (line 37-38) indicate that authors have no understanding of time analysis. If P. silasi is a sister taxon of P. icarii, then there should be only one TMRCA (time to the most recent common ancestor) and not two.
In the first line of introduction (line 42) authors mentions the year of genus description as 1940 should be 1840. Based on the nomenclatural rules, while mentioning Pangasius silasi the author name Dwivedi et al. 2017 cannot be in parenthesis as this is the original combination in which the species was described. Statements in lines 51-55 are assuming that Hora’s Satpura hypothesis is correct. There are several studies, including those based on genetics, which refute the Satpura hypothesis and authors should provide reference to these studies as well. Last statement in introduction (line 61-63) makes little sense. First, what is a timeline investigation? Second, how can a phylogeny based on distance based NJ method provide evolutionary status of the species, when NJ does not account for homoplasy or rate of nucleotide substitutions?
For morphometric data authors mention that they took characters as %TL, while in the diagnosis and comparison authors use %SL and same in the case of Table 1. Also, authors mention that characters are expressed as %HL when in Table 1 all characters are %SL. In general, subunits of head are expressed as %HL. Authors should provide the primer sequence or at least names of the primers and just referring to the article is not enough. Authors should include sequences for Pangasius macronema, P. conchophilus, P. elongatus and P. djambal from NCBI database. For NJ analysis authors do not provide any data on the distance method used for analysis. Authors mention Catla catla (correct species name Labeo catla) as outgroup although it is not present in the figure. Authors mention that they used HKY+G model of substitution for Bayesian analysis without providing any rationale for why this model was selected. Also, this entire analysis is erroneous. Authors should understand that COI is a protein coding gene so the three codon positions can have different rates of substitution. Authors should first partition the data in three codons and then use partition finder to find right partition scheme and nucleotide substitution models for the final partitions for doing ML and Bayesian analysis. The entire time tree analysis is erroneous. Authors mention that calibration point was used but no such point is provided. Also, one calibration point makes sense only if the strict clock hypothesis is supported. So authors need to statistically check that hypothesis first. Also, authors cannot use NJ tree for time tree because NJ is a distance based method that does not account for homoplasy and nucleotide substitution rates. Authors need to use character based methods such as ML and Bayesian analysis with proper partition and nucleotide substitution rates for the partitions to get the tree that can be sued for time tree analysis. Line 125-126 abut partitioning should be in results and not methods.
Diagnosis is not provided properly. What authors provide is a comparison. A diagnosis is a set of characters that separates the species from all its congeners unambiguously. Comparison can be provided as a separate heading where authors can provide comparison of the species with all its congeners. Authors provide several characters that are overlapping. Such characters should be removed from comparisons and authors should provide only distinct character states for diagnosis and comparison. Description is highly laconic and authors have not described the characters properly. Meristic counts should be accompanied with frequency of the counts in specimens. Simple and branched rays should be clearly stated for all fins. For caudal fin, procurrent and principal rays need to be stated. Lateral body coloration should be mentioned. Naming the species after the same institute from where the work was done is a clear conflict of interest as this can clearly be seen as a deliberate attempt to please seniors in the institute. This is authors’ choice so I will not comment further on it. Nevertheless, the current name will be icari and the extra “i” at the end is erroneous. Authors can also just call it “icar” and mention it as noun in apposition. Line 227-228 about accession number should be in methods and not in results. Line 273-275, is a baseless statement. Line 278-280, is a baseless statement.

·

Basic reporting

BASIC REPORT
• -Do not put the name of the new species in the title. It should be better only the genus name.
• -line 50 Pangasius should be in italics
• -Replace figure 10 with a better resolution
• -Replace figure 11 with a better resolution
• Drill down better legend from figure 9, what does the scale bar mean? what analysis is used for the reconstruction?

Experimental design

EXPERIMENTAL DESIGN
• In the phylogenetic analysis, very distant organisms were added, for example, Cyprinus carpio. This addition can cause data saturation, and I have not found a test to assess whether the data are reliable for the phylogenetic reconstruction. I recommend adding a saturation analysis.
• Line 114: “The best model for the present dataset was HKY+G.” The authors need to explain the model test
• Line 130: The Neighbor join is a Fenetic analysis. It is not highly recommended for phylogenetic reconstruction. Could the authors better explain the choice?
• Line 131: “The timetree was computed using 1 calibration constraint. This analysis involved nucleotide sequences.”. what is the calibration point? I didn't understand how the author found the cladogenesis dates. Would you please explain better?
• Comparative material: The genus has 22 species, but several are widely distributed and introduced, according to the authors. So I think there is a problem with comparing only closely distributed species. I think it's necessary to discriminate the new species from the other congeners.
• Habitat and distribution: -I strongly recommend adding a distribution map.
• Line 273: “The P. icarii appears to be the youngest among the known species”: I think that the authors need to revise this conclusion. As P. icarii is the sister group of P. silasi, both have species the same age, as they arose in the same cladogenetic process.
• General about molecula clock: I'm not convinced of the molecular clock analysis. I think this one needs to be much more detailed. For example, I did not identify the calibration point or the rate of evolution of the COI. There is a strong argument that this dating significantly underestimates the date of each node. According to Timetree, Cyprinidae diverged to Siluriformes less than 15 My years ago. But this is not what indicates most articles point that this cladogenesis has to be more than 125 My (Santini et al. 2009; Near et al. 2012; Arroyave et al.2013, Melo et al. 2021).

Validity of the findings

The authors needs to improve molecular clock analysis and increase comparative material.

Additional comments

Thanks for submitting the article for review, I learned a lot from the text written very clearly and precisely. The quality of the uploaded files is excellent, except for figure 10 and 11. I have some remarks about the text, which could be even better with some additions. I recommend the acceptance with major revisions.

Best regards

Dr. Guilherme José da Costa Silva, Univeridade Santo Amaro, São Paulo, Brazil

Reviewer 3 ·

Basic reporting

no comment

Experimental design

no comment

Validity of the findings

The analyzes used are not robust enough to support the conclusions.

Additional comments

Manuscript presents relevant information and deserve to be published after revisions proposed here are made. All the comments and suggestions are in the word file. I have a concern that should be analyzed before the manuscript can be accepted for publication:
I strongly suggest to authors do not discuss diversification time in this paper, the analyzes and data are not robust enough. But, if the authors decide to deal with this issue, I recommend: (1) added at least more one nuclear marker; (2) perform the analysis using the BEAST; (3) use fossil for calibration points.

Annotated reviews are not available for download in order to protect the identity of reviewers who chose to remain anonymous.

---

## Round 0.2 · Major Revisions

Still needs substantial improvement on the basis of reviewers' earlier and new comments. Please seek help for English editing.

Reviewer 1 ·

Basic reporting

English needs thorough revision. For instance, line 22: " It can be distinguished from its all sister species...." is a wrong statement grammatically and scientifically. According to cladistics terminology a taxon will have only one sister species that is the closest relative. Authors should mention this as, "It can be distinguished from its congeners...". There are numerous such incidences throughout the text and in figure captions. Authors should take help for English revision.

The species name also has an error. It should be "icari" and not "icarii" the second "i" makes little sense from zoological nomenclature point of view. For species named after institute abbreviation, it is also possible to use "-ia" so, "icaria". Authors should consult a taxonomist and name the species properly.

Authors have revised most of the figures properly, however, figure 7 is not legible. There is a need to increase the font size. In figure 2 it is not clear why authors are providing photograph of "Pangasiodon hypothalamus", a species not even in the same genus as that of the focal taxa.

Experimental design

In the earlier review a suggestion was made to include Poisson tree process (PTP) in addition to ASAP to delimit species. In the rebuttal letter authors mention that they have performed PTP analysis in the revised MS. In fact, PTP is mentioned in the revised abstract but there is no further mention of PTP throughout the text. PTP is not explained in methods. There are not results based on PTP and there is no discussion based on the same.

Authors mention that HKY+G was the best model. However, they do not mention how the model test was performed. Also, COI is a protein coding gene. So its first, second and third codons might have different rates of substitutions. Authors should partition their data into three codon positions and use partition findet to find the best partition scheme and best nucleotide substitution models for the partitions.

Inter and Intra specific genetic distances should be mentioned for the Pangasius species, preferably as a table.

Validity of the findings

In line no. 238 authors mention "Phylogenetic analysis using MEGA X ..." is not way of reporting results. MEGA X is a software package that implements various phylogenetic methods and authors should mention the phylogenetic methods, like ML, MP, etc. and not software names. The software package is already mentioned in the methods section.

Additional comments

The revised manuscript is better but not a substantial improvement from the earlier draft. Authors should take the earlier comments made by the reviewers seriously and revise the manuscript properly. Authors should understand that revision of the manuscript is beneficial for themselves and not the anonymous reviewers. Authors should read relevant literature to understand the proper scientific terminology used for presenting taxonomic and molecular findings. Authors should seek help for English grammar.

---

## Round 0.3 · accepted · Accept

Congratulations!
Hope this work will enrich the Pangasius diversity study in Peninsular India.

Reviewer 1 ·

Basic reporting

No comment

Experimental design

No comment

Validity of the findings

No comment

Additional comments

Authors have made appropriate changes based on earlier comments.